# Transformers perform in-context learning through neural networks

## Abstract

Transformer based neural sequence models exhibit remarkable ability to do in-context learning. Given some training examples, a pre-trained model can make accurate predictions on a novel input. This paper studies why transformers can learn different types of function classes in context. We first show by construction that transformers implement approximate gradient descent on parameters of neural networks and provide an upper bound for number of heads, hidden dimension, and number of layers of the transformer. We also show that transformers can learn deep and narrow neural networks, which has better approximation capabilities compared to shallow and wide neural networks, using less resource. Our results move beyond linearity in terms of in-context learning instances and provide an understanding of why transformers can learn many types of function classes through the bridge of neural networks.

## 1 Introduction

In-context learning (ICL) is a phenomenon first observed in NLP problems where large language models like GPT-4 can make accurate predictions based on few prompts without any update on model parameters. People's understanding on in-context learning is still limited, how and why can neural sequence models learn in-context remain a black box. Previous work mainly focus on two perspectives of in-context learning. One perspective explores what function classes can transformers learn in-context(Garg et al., 2022), another explains why transformers can implement learning algorithms (Akyürek et al., 2022).

The paper tries to explain why transformer-based predictors can learn different function classes in-context. We interpret in-context learning of a function as learning implicit neural networks that approximate the function. Currently there are mainly two understandings of in-context learning, one is based on gradient descent(Von Oswald et al., 2023; Dai et al., 2022; Akyürek et al., 2022), and the other views it as Bayesian Inference(Xie et al., 2021). We adapt the former perspective to investigate the hypothesis that when trained properly, transformers can perform approximate gradient descent on parameters of neural networks without any parameter update or fine tuning, and these neural networks are approximators of different functions.

In Section 3, we prove by construction that a family of transformers, with a wide range of activation functions (not necessarily restricted to the commonly used ReLU), can implement a step of approximate gradient descent on the parameters of the neural networks. We start by investigating on 2-layer neural networks and then generalize it to the $n$-layer neural networks setting. Upper bound for the number of heads, hidden dimension as well as number of layers needed of the transformer is provided, among which the number of layers is presented in a recursive fashion for the $n$-layer neural networks setting.

In Section 4, we view neural networks as bridges for transformers to learn function classes in-context, and provide an analysis on the resources it cost for a transformer to approximate the same function classes through neural networks of different depths and widths. We showcase that for transformers to learn indicator functions in-context, 2-layer neural networks are not sufficient as bridges since it will cause the number of heads of the transformer to be unacceptably large (for certain function classes), while deeper and narrower neural networks which achieve the same approximation accuracy cost less resource (number of parameter matrices) of the transformer. We also present a condition on

when deep networks does better than shallow ones in terms of approximating smooth functions and requiring smaller transformer size.

## 1.1 RELATED WORK

**In-context learning** In-context learning has been studied both empirically and theoretically. Garg et al. (2022) empirically show that transformers can learn linear functions, two-layer ReLu neural networks and decision trees in-context. Min et al. (2022) study what aspects of demonstrations impact the performance of in-context learning. As for the theoretical part, Xie et al. (2021) explains ICL as implicit Bayesian inference, while Akyürek et al. (2022), Von Oswald et al. (2023) and Dai et al. (2022) all understand in-context learning as transformers performing gradient descent. These works all only focus on linear models or their variants without providing an error bound for gradient descent steps. A more recent work (Bai et al., 2023) also investigates gradient descent on more general functions, like 2-layer neural networks and demonstrates the model selection ability of transformers. We extend the their result on 2-layer neural networks to an $n$-layer neural networks setting and also provide a tighter bound on the number of heads required of the transformer.

**Neural networks and approximation theorems** People are interested in the the approximation abilities of neural networks. Many results have shown the universal approximation property of neural networks in approximating different function classes (Hornik et al., 1989; Hornik, 1991; Barron, 1993). While these universal approximation theorems focus on neural networks with certain depths, more recent work starts to explore the expressing power of deep neural networks due to their development and success. Yarotsky (2017); Liang and Srikant (2016) both show the approximation abilities of deep neural networks. Safran and Shamir (2017) shows the width-depth tradeoffs of neural networks by proving the inapproximability with 2-layer neural networks an the approximability of 3-layer neural networks in terms of approximating indicator functions.

## 2 PRELIMINARIES

## 2.1 TRANSFORMERS

A Transformer layer contains two sub-layers, the attention layer and the feed forward layer, which in essence is an MLP layer. We denote the input sequence to the transformer as $\mathbf{H} = [\mathbf{h}_1, \cdots, \mathbf{h}_N] \in \mathbb{R}^{D \times N}$.

**Definition 1.** *(Attention layer) An attention layer with $M$ heads is denoted as $\mathrm{Attn}_{\boldsymbol{\theta}}(\cdot)$ where $\boldsymbol{\theta} = \{\mathbf{V}_m, \mathbf{Q}_m, \mathbf{K}_m\}_{m \in [M]}$. The output of this layer on the input matrix $\mathbf{H}$ is:*

$$\mathrm{Attn}_{\boldsymbol{\theta}}(\mathbf{H}) = \mathbf{H} + \frac{1}{N} \sum_{m=1}^{M} (\mathbf{V}_m \mathbf{H}) \times \sigma(\mathbf{Q}_m \mathbf{H})^{\top} (\mathbf{K}_m \mathbf{H}))$$

*where $\sigma$ is an activation function (not necessarily restricted to the ReLU function). For each column:*

$$[\mathrm{Attn}_{\boldsymbol{\theta}}(\mathbf{H})]_i = \mathbf{h}_i + \frac{1}{N} \sum_{m=1}^{M} \sum_{j=1}^{N} \sigma\left(\langle \mathbf{Q}_m \mathbf{h}_i, \mathbf{K}_m \mathbf{h}_j \rangle\right) \cdot \mathbf{V}_m \mathbf{h}_j.$$

**Definition 2.** *(MLP layer) An MLP layer with hidden dimension $D'$ is denoted as $\mathrm{MLP}_{\boldsymbol{\theta}}(\cdot)$ where $\boldsymbol{\theta} = (\mathbf{W}_1, \mathbf{W}_2) \in \mathbb{R}^{D' \times D} \times \mathbb{R}^{D \times D'}$. The output of this layer on input $\mathbf{H}$ is*

$$\mathrm{MLP}_{\boldsymbol{\theta}}(\mathbf{H}) = \mathbf{H} + \mathbf{W}_2 \sigma(\mathbf{W}_1 \mathbf{H})$$

*where $\sigma$ is an activation function. For each column:*

$$[\mathrm{MLP}_{\boldsymbol{\theta}}(\mathbf{H})]_i = \mathbf{h}_i + \mathbf{W}_2 \sigma(\mathbf{W}_1 \mathbf{h}_i).$$

We doesn't require the transformer activation function to be restricted to a specific type, and next we give a definition on the class of activation functions we focus on.

**Definition 3.** *(General decay condition) We call an activation function $\sigma$ satisfies the general decay condition if $\sigma \in W_{loc}^{m,\infty}(\mathbb{R})$ is non-zero and there exists a $\nu \in \{\sum_{i=1}^{n} \beta_i \sigma(\omega_i \cdot x + b_i) : \omega_i, b_i, \beta_i \in \mathbb{R}\}$ which satisfies*

$$|\nu^{(k)}(t)| \le C_p (1 + |t|)^{-p}$$

*for $0 \le k \le m$ and some $p > 1$.*

We briefly note that here $W^{m,\infty}$ denotes the Sobolev space, and most common activation functions do satisfy the general decay condition (Siegel and Xu, 2020).

## 2.2 Neural Networks

We formulate the mathematical representation of an $n$-layer neural network as below:

**Definition 4.** *(n-layer neural networks)* *We denote the output of an $n$-layer neural network on the input $x \in \mathbb{R}^d$ as*

$$\text{pred}_n(\mathbf{w}, \mathbf{x}) = \mathbf{W}^{(n)}(\text{r}(\mathbf{W}^{(n-1)}(\text{r}(\cdots \text{r}(\mathbf{W}^{(1)}\mathbf{x})))))$$

*where $r$ is an activation function and $\mathbf{w} = (\mathbf{W}^{(1)}, \cdots, \mathbf{W}^{(n)})$, $\mathbf{W}^{(i)} \in \mathbb{R}^{K_i \times K_{i-1}}$ for $i = 1, \cdots, n$ with $K_n = 1, K_0 = d$. We denote the $k$-th row vector of the matrices $\mathbf{W}^{(i)} (i \in [n-1])$ as $\mathbf{v}_{i,k}$, and the $k$-th element in the vector $\mathbf{W}^{(n)}$ as $u_k$.*

In the $n$-layer neural network setting above, we omit the bias terms and let the output be a number instead of a vector for simplicity. Also, the activation function $r$ act on each element of the vector.

## 2.3 In-context learning

Here we introduce our in-context learning (ICL) setting. The training examples (prompts) are sampled from a distribution P, denoted as $\mathcal{D} = (\mathbf{x}_i, y_i)_{i \in [N]}$, and a novel input $\mathbf{x}_{N+1}$ is sampled from $P_{\mathbf{x}}$. So each instance is of the form $(\mathcal{D}, \mathbf{x}_{N+1})$. Here $\mathbf{x}_i \in \mathbb{R}^d$.

More specifically, we denote the input to the transformer as

$$\mathbf{H} = \begin{bmatrix} \mathbf{x}_1, & \mathbf{x}_2, \cdots, & \mathbf{x}_N, & \mathbf{x}_{N+1} \\ y_1, & y_2, \cdots, & y_N, & 0 \\ \mathbf{p}_1, & \mathbf{p}_2, \cdots, & \mathbf{p}_N, & \mathbf{p}_{N+1} \end{bmatrix} \in \mathbb{R}^{D \times (N+1)}$$

where $\mathbf{p}_i$ are vectors in hidden space of the form

$$\mathbf{p}_i = \begin{bmatrix} \mathbf{0}_{D-d-3} \\ 1 \\ 1\{i < N+1\} \end{bmatrix}$$

A transformer takes the prompt input $\mathbf{H}$ and makes a prediction on the label corresponding to $\mathbf{x}_{N+1}$. The prediction $\hat{y}_{N+1}$ is stored in the output matrix $\tilde{\mathbf{H}}$ in the position next to $y_N$.

## 3 Gradient Descent on multi-layer neural network

We begin our analysis on 2-layer neural networks, and then generalize it to the $n$-layer networks setting. To perform gradient descent on the neural networks, we consider the following optimization problem on the loss function:

$$\min_{\mathbf{w} \in \mathcal{W}} L_N(\mathbf{w}) = \frac{1}{2N} \sum_{i=1}^{N} l(\text{pred}(\mathbf{x}_i; \mathbf{w}), y_i)$$

Now we present necessary assumptions begin our analysis.

Following Barron (1993), We define $\mathcal{B}_s$ to be the space of functions $f : \mathbb{R}^d \to \mathbb{R}$ with bounded Barron norm:

$$\|f\|_{\mathcal{B}^s} = \int_{\mathbb{R}^d} (1 + |\boldsymbol{\omega}|)^s |\hat{f}(\boldsymbol{\omega})| d\boldsymbol{\omega}.$$

**Assumption 1.** *Both the activation function $r$ in neural networks and the loss function $l$ has finite Barron norm.*

As discussed in Barron (1993), $B_s$ is closed to multiplication, linear combination and translation. Also, sigmoidal functions, functions with derivatives of sufficiently high order and Boolean functions are all in $B_s$, so this should include most common activation functions and loss functions in practice. We also point out that in Bai et al. (2023) they used the assumption that $r, l$ are both $C^4$ smooth, which is a stricter assumption since $C^4$ functions do have bounded Barron norm.

During the process of gradient descent, it is likely that the neural networks parameter $\mathbf{w}$ goes out of its domain $\mathcal{W}$, so we need the following assumption to project $\mathbf{w}$ onto $\mathcal{W}$.

**Assumption 2.** $\mathcal{W}$ *as the domain of* $\mathbf{w}$ *is compact and there exists some MLP layer parameter such that the MLP layer projects* $\mathbf{w}$ *to* $\mathcal{W}$.

### 3.1 GRADIENT DESCENT ON 2-LAYER NEURAL NETWORKS

A 2-layer neural network can be written as

$$\mathrm{pred}_2(\mathbf{w}, \mathbf{x}) = \mathbf{W}^{(2)}(\mathrm{r}(\mathbf{W}^{(1)}\mathbf{x}))$$

We want to show that transformers can implement gradient descent on 2-layer neural networks in-context without any parameter update.

We note that

$$\nabla_{\mathbf{w}} L_N(\mathbf{w}) = \frac{1}{N}\sum_{i=1}^{N} \partial_1 l(\mathrm{pred}(\mathbf{x_i}; \mathbf{w}), \mathrm{y_i}) \cdot \nabla_{\mathbf{w}}\mathrm{pred}(\mathbf{x_i}; \mathbf{w}),$$

where $\partial_1 l$ is the partial derivative of $l$ with respect to the first component. Furthermore,

$$\nabla_{\mathbf{w}}\mathrm{pred}(\mathbf{x_i}; \mathbf{w}) = \begin{bmatrix} u_1 \cdot r'(\langle \mathbf{v}_1, \mathbf{x}_i\rangle) \cdot \mathbf{x}_i \\ r(\langle \mathbf{v}_1, \mathbf{x}_i\rangle) \\ \vdots \\ u_K \cdot r'(\langle \mathbf{v}_K, \mathbf{x}_i\rangle) \cdot \mathbf{x}_i \\ r(\langle \mathbf{v}_k, \mathbf{x}_i\rangle) \end{bmatrix}$$

We show below that a 2-layer transformer can compute one step of approximate gradient descent, following the intuition of Siegel and Xu (2020): The first self attention sub-layer computes and stores approximate $\mathrm{pred}(\mathbf{x_i}; \mathbf{w})$ in hidden space, and the MLP sub-layer computes and stores $\partial_1 l$, while the second attention sub-layer computes and stores $\mathbf{w} - \eta \nabla L_N(\mathbf{w})$, and the last MLP sub-layer maps this result of one step gradient descent to the domain of $\mathbf{w}$.

**Theorem 1** (ICGD on 2-layer NNs). *Under* **Assumption 1** *and* **Assumption 2**, *there exists a family of 2-layer transformers (with activation functions satisfying the general decay condition) such that for any input data* $(\mathcal{D}, \mathbf{x}_{N+1})$ *and any* $\mathbf{w}$, *a transformer performs approximate gradient descent on the neural networks parameter* $\mathbf{w}$:

$$\mathbf{w}_\eta^+ = \mathrm{Proj}_{\mathcal{W}}(\mathbf{w} - \eta \nabla \mathrm{L_N}(\mathbf{w}) + \epsilon(\mathbf{w})), \quad \|\epsilon(\mathbf{w})\| \le \eta\epsilon.$$

*Furthermore, the upper bound for number of heads and hidden dimension for the transformer is:*

$$\max_{l \in [2]} M^{(l)} \le \mathcal{O}(\epsilon^{-2}), \quad \max_{l \in [2]} D^{(l)} \le \mathcal{O}(\epsilon^{-2})$$

**Remark 1.** *We note here that in the theorem when we say a transformer performs approximate gradient descent on the neural networks parameter, we mean that the transformer maps each column of the input matrix* $\mathbf{H}$: $\mathbf{h}_i = [\mathbf{x}_i; y_i'; \mathbf{w}; \mathbf{0}; 1; t_i]$ *to the output vector* $\mathbf{h}_i' = [\mathbf{x}_i; y_i'; \mathbf{w}_\eta^+; \mathbf{0}; 1; t_i]$, *where only the parameter* $\mathbf{w}$ *is updated according to gradient descent and the other parts remain unchanged.*

### 3.2 GRADIENT DESCENT ON $n$-LAYER NEURAL NETWORKS

As in Definition 4, $n$-layer neural networks can be formulated as

$$\mathrm{pred}_n(\mathbf{w}, \mathbf{x}) = \mathbf{W}^{(n)}(\mathrm{r}(\mathbf{W}^{(n-1)}(\mathrm{r}(\cdots \mathrm{r}(\mathbf{W}^{(1)}\mathbf{x})))))$$

We theoretically prove that transformers can implement gradient descent on $n$-layer neural networks in context and provide an upper bound for number of heads, hidden dimension of transformers as well as give a recurrence relation on the number of transformer layers.

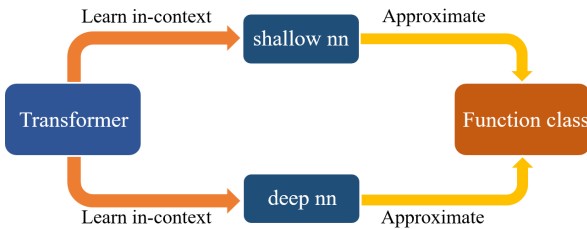

Figure 1: Neural Networks as bridges for transformers to learn function classes in-context. Though transformers can learn both deep and shallow neural networks in-context, learning them requires different transformer sizes.

**Theorem 2** (ICGD on $n$-layer NNs). *Under **Assumption 1** and **Assumption 2**, there exists a family of $a_n$-layer transformers (with activation functions satisfying the general decay condition) such that for any input data $(\mathcal{D}, \mathbf{x}_{N+1})$ and any $\mathbf{w}$, a transformer performs approximate gradient descent on the $n$-layer neural networks parameter $\mathbf{w}$:*

$$\mathbf{w}_\eta^+ = \mathrm{Proj}_{\mathcal{W}}(\mathbf{w} - \eta\nabla\mathrm{L}_{\mathrm{N}}(\mathbf{w}) + \epsilon(\mathbf{w})), \quad \|\epsilon(\mathbf{w})\| \leq \eta\epsilon.$$

*where $a_n$ satisfies $\mathcal{O}(a_n) = \mathcal{O}(n) + \mathcal{O}(a_{n-1})$. Furthermore, the upper bound for number of heads and hidden dimension for the transformer is :*

$$\max_{l \in [a_n]} M^{(l)} \leq \mathcal{O}(nK^2\epsilon^{-2}), \quad \max_{l \in [a_n]} D^{(l)} \leq \mathcal{O}(nK^2\epsilon^{-2})$$

*where $K$ denotes the maximum width of the neural networks: $K = \max\{K_0, K_1, \cdots, K_n\}$.*

**Remark 2.** *We note that $a_n$, the number of transformer layers required in the above theorem is of order $\mathcal{O}(n^2)$, which is a decent growth rate considering the fast growth of neural networks neurons.*

**Trade-off of width and depth of neural networks** We note here that people are interested in the approximation capabilities of neural networks, thus it's worthwhile discussing how the trade-off between width and depth of neural networks can affect the approximation error of the gradient descent step. If we control the error in the approximate gradient descent to be $\epsilon$, then in order for the transformer to learn two different neural networks with the same magnitude of number of heads, we require $nK^2$ to be a constant. Thus controlling the width of the network is more efficient than controlling the depth of the neural networks in terms of maintaing the same approximation error.

## 4 NEURAL NETWORKS AS BRIDGES FOR TRANSFORMER TO LEARN FUNCTION CLASSES IN-CONTEXT

Since neural networks are universal approximators, and transformers can learn neural networks in context, a natural question is whether transformers can learn function classes that neural networks can approximate. Our theorem bridges the gap between transformers and neural networks, and previous work on approximation theorems of neural networks has bridged the gap between neural networks and function classes, so it is natural to consider the whole path of transformers learning function classes in-context as shown in Fig. 1. But how much resources does it cost for a transformer (number of layers of the transformer and number of heads for each attention layer, or equivalently number of parameter matrices) to approximate a neural network (as an approximator)? We consider two types of function classes: the indicator functions of $L_2$ balls and smooth functions.

### 4.1 INDICATOR FUNCTIONS OF $L_2$ BALLS

Safran and Shamir (2017) proves the inapproximability of 2-layer neural networks and the approximability of 3-layer neural networks. We present them as lemmas below.

**Lemma 1.** *The following holds for some positive univeral constants $c_1, c_2, c_3, c_4$, and any network employing an activation functioin satisfying Assumptions 1 and 2 in Eldan and Shamir (2016): For any $d > c_1$, and any non-singular matrix $A \in \mathbb{R}^{d\times d}, \mathbf{b} \in \mathbb{R}^d$ and $r \in (0, \infty)$, there exists a*

*continuous probability distribution $\gamma$ on $\mathbb{R}^d$, such that for any function $g$ computed by a 2-layer network of width at most $c_2 \exp(c_4 d)$, and for the function $f(\mathbf{x}) = \mathbf{1}(\|A\mathbf{x} + \mathbf{b}\| \leq r)$, we have*

$$\int_{\mathbb{R}^d} (f(\mathbf{x}) - g(\mathbf{x}))^2 \cdot \gamma(\mathbf{x}) d\mathbf{x} \geq \frac{c_2}{d^4}.$$

**Lemma 2.** *Given $\delta > 0$, for any activation function $\sigma$ satisfying Assumption 1 in Eldan and Shamir (2016) and any continuous probability distribution $\mu$ on $\mathbb{R}^d$, there exists a constant $c_\sigma$ dependent only on $\sigma$, and a function $g$ expressible by a 3-layer network of width at most $\max\{8c_\sigma d^2/\sigma, c_\sigma \sqrt{1/2\delta}\}$, such that the following holds:*

$$\int_{\mathbb{R}^d} (g(\mathbf{x}) - \mathbf{1}(\|\mathbf{x}\|_2 \leq 1))^2 \mu(\mathbf{x}) d\mathbf{x} \leq \delta,$$

*where $c_\sigma$ is a constant depending solely on $\sigma$.*

These two lemmas reveal that the indicator of the $L^2$ ball can be better approximated by a 3-layer neural network with width $\mathcal{O}(d^2)$ than a 2-layer neural network requiring width at least exponential in the input dimension. Now using these lemmas we obtain the following theorem on transformer resource it takes to learn these two networks.

**Theorem 3.** *Let $f(\mathbf{x}) = \mathbf{1}(\|A\mathbf{x} + \mathbf{b}\| \leq r)$ be the indicator of unit ball. Consider two neural networks $\mathrm{NN_A}$ and $\mathrm{NN_B}$ that approximates the function $f$ with error $\epsilon \in (0, 1)$, where $\mathrm{NN_A}$ is a 2-layer, width $\mathcal{O}(\exp(\epsilon^{-1/4}))$ and $\mathrm{NN_B}$ is a 3-layer, width $\mathcal{O}(\epsilon^{-1})$. It takes two transformers $\mathrm{TF_A}, \mathrm{TF_B}$ to learn these two neural networks. $\mathrm{TF_A}$ needs $\mathcal{O}(\exp(2\epsilon^{-1/4}))$ parameter matrices and $\mathrm{TF_B}$ needs $\mathcal{O}(\epsilon^{-1})$ parameter matrices.*

*Proof.* Recall Theorem 2 and we know that the number of transformer layers required is $\mathcal{O}(nK^2)$ (Here we the gradient error $\epsilon$ is absorbed in the $\mathcal{O}$ notation since it is a fixed constant independent of the networks approximation error). So the transformer size is determined by number of layers times number of heads, which is $\mathcal{O}(n^2 K^2)$. Now bring in the width of $\mathrm{NN_A}$ and $\mathrm{NN_B}$ immediately yields the results, concluding the proof. $\qquad\square$

The theorem points out the exponential explosion in transformer size if a 2-layer neural network is learned to approximate the indicator. However, a 3-layer neural network requires much smaller transformer size. This showcases the neccessity of bringing deeper neural networks into consideration.

## 4.2 Smooth functions

We use the approximation results of deep neural networks achieved by Yarotsky (2017).

**Lemma 3.** *For any $d, n$ and $\epsilon \in (0, 1)$, there is a ReLU network architecture that*

- *is capable of expressing any function from $F_{d,n}$ with error $\epsilon$;*

- *has the depth at most $c(\ln(1/\epsilon) + 1)$ and at most $c\epsilon^{-d/n}(\ln(1/\epsilon) + 1)$ computation units, with some constant $c = c(d, n)$.*

**Lemma 4.** *Let $f \in C^2([0, 1]^d)$ be a nonlinear function. Then, for any fixed $L$, a depth-$L$ ReLU network approximating with error $\epsilon \in (0, 1)$ must have at least $c\epsilon^{-1/(2(L-1))}$ computation units, with some constant $c = c(f, L) > 0$.*

While Lemma 3 gives the upper bound for a relatively deep neural network, Lemma 4 show the slow approximation of smooth functions by shallow networks.

Below we state a formal theorem for the resource of a transformer it takes to learn two neural network: one is deep and narrow, another is shallow and wide.

**Theorem 4.** *For any $d$ and $n > 2$, let $f \in \mathcal{W}^{n,\infty}([0, 1]^d)$. Consider two neural networks $\mathrm{NN_A}$ and $\mathrm{NN_B}$ that approximates the function $f$ with error $\epsilon \in (0, 1)$. $\mathrm{NN_A}$ has depth $\mathcal{O}(\ln(1/\epsilon) + 1)$ and width $\mathcal{O}(\epsilon^{-d/n})$ as in Lemma 3, and $\mathrm{NN_B}$ has depth $L$ and width $\mathcal{O}(\epsilon^{-1/(2(L-1))})$ as in Lemma 4. We also have two transformers, $\mathrm{TF_A}$ and $\mathrm{TF_B}$, pre-trained to learn these neural networks. If we fix the accuracy of gradient descent in Theorem 2, then $\mathrm{TF_A}$ needs $\mathcal{O}((\ln(1/\epsilon)^3 \epsilon^{-2d/n}))$ parameter matrices and $\mathrm{TF_B}$ needs $\mathcal{O}(\epsilon^{-1/(L-1)})$ parameter matrices. In particular, if $2(L-1)d < n$, then it costs a transformer less resource to train neural network $A$ than $B$.*

This theorem can be proved in exactly the same way as Theorem 3 thus we ommit the proof. Intuitively, the smoother the function is and the lower dimension the input $\mathbf{x}$ is, the better deep neural networks perform in terms of approximation and transformer resource cost.

## 5 CONCLUSION

We provide results on transformers learning $n$-layer neural networks through gradient descent and view in-context learning of a function as the process of learning neural networks which approximate this function. We also emphasize that our results gives a theoretical bound on the size of the transformer required to implement in-context learning.

Our work shed some light on deeper understanding of in-context learning. In fact, our results suggest that transformers can in-context learn any function classes that can be approximated by neural networks this way. We also present the resource required for a transformer to learn indicator functions and smooth functions (in particular $C^n$ functions), showing 2-layer neural networks aren't sufficient for in-context learning due to the explosion of transformer size. Also, the smoother the function is and the lower dimension the input is the better deep neural networks perform in terms of transformer resource cost. We believe our work brings a new perspective to the understanding of in-context learning and opens up new directions for empirical exploration of in-context learning.

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
