## A  THEOREM 2

We provide the proof of Theorem 2 here. We note that Theorem 1 is a special case of Theorem 2 thus we only provide the proof of the more genral case. We provide Theorem 1 separately for two reasons: first it is the most simple and empirically verified case (transformers can learn 2-layer neural networks in-context), second it is the initial condition for recursion of the general case in terms of transformer layers.

First we need an approximation theorem of neural networks (Siegel and Xu (2020) corollary 1). We denote $\Sigma_d^n(\sigma) = \{\sum_{i=1}^n \beta_i \sigma(\boldsymbol{\omega}_i \cdot \mathbf{x} + b_i) : \omega_i \in \mathbb{R}^d, b_i, \beta_i \in \mathbb{R}\}$. We also define $\mathcal{B}_s$ to be the space of functions $f : \mathbb{R}^d \to \mathbb{R}$ with bounded Barron norm

$$\|f\|_{\mathcal{B}^s} = \int_{\mathbb{R}^d} (1 + |\boldsymbol{\omega}|)^s |\hat{f}(\boldsymbol{\omega})| d\boldsymbol{\omega}.$$

**Lemma 5.** *Let $\Omega \subset \mathbb{R}^d$ be a bounded domain. If the activation function $\sigma \in W^{m,\infty}_{loc}(\mathbb{R})$ is non-zero and there exists a $\nu \in \Sigma^{n_0}_1(\sigma)$ which satisfies the polynomial decay condition*

$$|\nu^{(k)}(t)| \leq C_p(1 + |t|)^{-p}$$

*for $0 \leq k \leq m$ and some $p > 1$, we have*

$$\inf_{f_n \in \Sigma^n_d(\sigma)} \|f - f_n\|_{H^m(\Omega)} \leq |\Omega|^{\frac{1}{2}} \sqrt{n_0} C(p, m, dim(\Omega), \sigma) n^{-\frac{1}{2}} \|f\|_{\mathcal{B}^{m+1}}$$

*for any $f \in \mathcal{B}^{m+1}$.*

Note that when $m = 0$, the Sobolev space $H^0$ is in effect the $L^2$ space, and we set $m = 0$ in the proof of Theorem 1.

*Proof.* We first observe that $\nu \in \Sigma^{n_0}_1(\sigma)$ implies that

$$\Sigma^n_d(\nu) \subset \Sigma^{nn_0}_d(\sigma)$$

So we only need to prove that the result without the $\sqrt{n_0}$ term holds for $\sigma$ satisfying the polynomial decay condition itself.

The decay condition implies that $\sigma \in L^1(\mathbb{R})$ and thus the Fourier transform of $\sigma$ is well-defined. Since $\sigma \neq 0$, we have

$$0 \neq \hat{\sigma}(a) = \frac{1}{2\pi} \int_{\mathbb{R}} \sigma(\boldsymbol{\omega} \cdot \mathbf{x} + b) e^{-ia(\boldsymbol{\omega} \cdot \mathbf{x} + b)} db$$

so we have

$$e^{ia\omega \cdot x} = \frac{1}{2\pi\hat{\sigma}(a)} \int_{\mathbb{R}} \sigma(\omega \cdot x + b) e^{-iab} db$$

Thus

$$f(x) = \int_{\mathbb{R}^d} e^{i\omega x} \hat{f}(\omega) d\omega$$
$$= \int_{\mathbb{R}^d} \int_{\mathbb{R}} \frac{1}{2\pi\hat{\sigma}(a)} \sigma(\frac{\omega}{a} x + a) \hat{f}(\omega) e^{-iab} db d\omega.$$

The above integral is on an unbounded domain, but the decay assumption on the Fourier transform of $f$ allows us to normalize the integral in the $\omega$ deriction. By the triangle inequality and the boundedness of $x \in \Omega$, we have

$$|\frac{\omega}{a} \cdot x + b| \geq \max(0, |b| - \frac{R|\omega|}{|a|}).$$

where $R$ is the maximum norm of an element of $\Omega$. WLOG, we can translate $\Omega$ so that it contains the origin and $R \leq \operatorname{diam}(\Omega)$. Combining this with the polynomial decay of $\omega$ implies that

$$|\sigma^{(k)}(\frac{\omega}{a} \cdot x + b)| \leq C_p(1 + |\frac{\omega}{a} \cdot x + b|)^{-p}$$
$$\leq C_p(1 + \max(0, |b| - \frac{R|\omega|}{|a|}))^{-p}.$$

Thus the function $h$ defined by

$$h(b, \omega) = (1 + \max(0, |b| - \frac{R|\omega|}{|a|}))^{-p}$$

provides an upper bound on $\sigma^{(k)}(\frac{\omega}{a} \cdot x + b)$ uniformly in $x$. Moreover, we calculate that

$$\int_{\mathbb{R}} h(b, \omega) db = \int_{|b| \leq \frac{R|\omega|}{|a|}} db + 2 \int_{b > \frac{R|\omega|}{|a|}} (1 + b - \frac{R|\omega|}{|a|})^{-p} db$$
$$= 2R|a|^{-1}|\omega| + 2[(1 - p)^{-1} \times (1 + b - \frac{R|\omega|}{|a|})^{1-p}]^{\infty}_{\frac{R|\omega|}{|a|}}$$
$$= 2R|a|^{-1}|\omega| + \frac{2}{p - 1}$$
$$\leq C_1(p, \operatorname{diam}(\Omega), \sigma)(1 + |\omega|).$$

Combining the above with our assumption on the Fourier transform we get

$$I(p, \Omega, \sigma, f) = \int_{\mathbb{R}^d} \int_{\mathbb{R}} (1 + |\omega|)^m h(b, \omega) |\hat{f}(\omega)| db d\omega \tag{1}$$

$$\leq C_1(p, \text{diam}(\Omega), \sigma) \|f\|_{\mathcal{B}^{m+1}} \tag{2}$$

Now we use this to introduce a probability measure $\lambda$ on $\mathbb{R}^{d+1}$ given by

$$d\lambda = \frac{1}{I(p, \Omega, \sigma, f)} (1 + |\omega|)^m h(b, \omega) |\hat{f}(\omega)| db d\omega,$$

using this we write

$$f(x) = \mathbb{E}_{d\lambda}(J(\omega, b) e^{i\theta(\omega, b)} \sigma(\frac{\omega}{a} x + b)),$$

where

$$\theta(\omega, b) = \theta(\hat{f}(\omega)) - \theta(\hat{\sigma}(a)) - ab$$

and

$$J(\omega, b) = (2\pi |\hat{\sigma}(a)|)^{-1} I(p, \Omega, \sigma, f)(1 + |\omega|)^{-m} h(b, \omega)^{-1}.$$

We denote the real part of $e^{i\theta(\omega, b)}$ as $\chi(\omega, b) \in [-1, 1]$, then we have

$$f(x) = \mathbb{E}_{d\lambda}(J(\omega, b) \chi(\omega, b) \sigma(\frac{\omega}{a} x + b)).$$

We denote $f(x) = \mathbb{E}_{d\lambda}(f_{\omega, b}(x))$. Then we use Lemma 1 from Barron (1993) to conclude that for each $n$ there exists an $f_n$ which is a convex combination of at most $n$ distinct $f_{\omega, b}$, and thus $f_n \in \Sigma_d^n(\sigma)$, such that

$$\|f - f_n\|_{H^m(\Omega)} \leq C n^{-\frac{1}{2}}, \tag{3}$$

where $C = \sup_{\omega, b} \|f_{\omega, b}\|_{H^m(\Omega)}$. Now, since $\Omega$ is bounded, it has finite measure, and we use Cauchy-Schwartz to get

$$\|f_{\omega, b}\|_{H^m(\Omega)}\| \leq |\Omega|^{\frac{1}{2}} \|f_{\omega, b}\|_{W^{m, \infty}(\Omega)},$$

so we only need to bound $\|D_x^\alpha f_{\omega, b}\|_{L^\infty(\Omega)}$ for each $|\alpha| \leq m$.

$$\|D_x^\alpha f_{\omega, b}\|_{L^\infty(\Omega)} \leq \|J(\omega, b) D_x^\alpha \sigma(a^{-1} \omega x + b)\|_{L^\infty(\Omega)}$$

$$\leq (2\pi |a^{|\alpha|} \hat{\sigma}(a)|)^{-1} I(p, \Omega, \sigma, f)(1 + |\omega|)^{-m}) \times \|h(b, \omega)^{-1} D_x^\alpha \sigma(a^{-1} \omega x + b)\|_{L^\infty(\Omega)}.$$

Since $|\alpha| \leq m, \sigma \in W^{m, \infty}$, we have

$$|D_x^\alpha \sigma(a^{-1} \omega x + b)| \leq |a|^{-|\alpha|} (1 + |\omega|)^m \sigma^{(|\alpha|)}(a^{-1} \omega x + b)$$

So we get

$$\|D_x^\alpha f_{\omega, b}\|_{L^\infty(\Omega)} \leq (2\pi |a^{|\alpha|} \hat{\sigma}(a)|)^{-1} I(p, \Omega, \sigma, f) \times \|h(b, \omega)^{-1} \sigma^{(|\alpha|)}(a^{-1} \omega x + b)\|_{L^\infty(\Omega)}.$$

What's more we have

$$\|h(b, \omega)^{-1} \sigma^{(|\alpha|)}(a^{-1} \omega x + b)\|_{L^\infty(\Omega)} \leq C_p.$$

So we obtain

$$\sup_{\omega, b} \|f_{\omega, b}\|_{H^m(\Omega)} \leq |\Omega|^{\frac{1}{2}} (2\pi |\hat{\sigma}(a)|)^{-1} I(p, \Omega, \sigma, f) C_p \sum_{|\alpha| \leq m} |a|^{-|\alpha|}$$

Finally using Eqs. (1) and (3) we get

$$\inf_{f_n \in \Sigma_d^n(\sigma)} \|f - f_n\|_{H^m(\Omega)} \leq |\Omega|^{\frac{1}{2}} \sqrt{n_0} C(p, m, \text{dim}(\Omega), \sigma) n^{-\frac{1}{2}} \|f\|_{\mathcal{B}^{m+1}}.$$

$\square$

Now we are ready to give a proof of Theorem 2.

**Theorem 2** (ICGD on $n$-layer NNs). *Under **Assumption 1** and **Assumption 2**, there exists a family of $a_n$-layer transformers (with activation functions satisfying the general decay condition) such that for any input data $(\mathcal{D}, \mathbf{x}_{N+1})$ and any $\mathbf{w}$, a transformer performs approximate gradient descent on the $n$-layer neural networks parameter $\mathbf{w}$:*

$$\mathbf{w}_\eta^+ = \text{Proj}_{\mathcal{W}}(\mathbf{w} - \eta \nabla L_N(\mathbf{w}) + \epsilon(\mathbf{w})), \quad \|\epsilon(\mathbf{w})\| \leq \eta \epsilon.$$

*where $a_n$ satisfies $\mathcal{O}(a_n) = \mathcal{O}(n) + \mathcal{O}(a_{n-1})$. Furthermore, the upper bound for number of heads and hidden dimension for the transformer is :*

$$\max_{l \in [a_n]} M^{(l)} \leq \mathcal{O}(nK^2 \epsilon^{-2}), \quad \max_{l \in [a_n]} D^{(l)} \leq \mathcal{O}(nK^2 \epsilon^{-2})$$

*where $K$ denotes the maximum width of the neural networks: $K = \max\{K_0, K_1, \cdots, K_n\}$.*

*Proof.* We note that

$$\nabla_{\mathbf{w}} L_N(\mathbf{w}) = \frac{1}{N} \sum_{i=1}^N \partial_1 l(\text{pred}(\mathbf{x}_i; \mathbf{w}), y_i) \cdot \nabla_{\mathbf{w}} \text{pred}(\mathbf{x}_i; \mathbf{w}), \tag{4}$$

where $\partial_1 l$ is the partial derivative of $l$ with respect to the first component. Moreover we have

$$\nabla_{\mathbf{v}_{i,j}} \text{pred}(\mathbf{x}; \mathbf{w}) = \sum_{k=1}^{K_n} u_k r'(\mathbf{v}_{n-1,k}^\top r(\mathbf{W}^{(n-2)} r(\cdots))) \nabla_{\mathbf{v}_{i,j}} \mathbf{v}_{n-1,k}^\top r(\mathbf{W}^{(n-2)} r(\cdots))$$

Now we use Lemma 3 to approximate the functions $r(t), \partial_1 l(t, y)$ and $s \cdot r'(t)$. Note that in the following expressions we denote $\boldsymbol{\omega}_i \cdot \mathbf{x} + b_i$ as $\langle \mathbf{a}_i, [\mathbf{x}; 1] \rangle$.

- The function $r(t)$ is approximated by $\bar{r}(t)$ on $[-R_1, R_1]$:

$$\bar{r}(t) = \sum_{m=1}^{M_1} \beta_m^1 \sigma(\langle \mathbf{a}_m^1, [t; 1] \rangle) \quad \text{with} \quad M_1 \leq \mathcal{O}(\epsilon_r^{-2})$$

  such that $\|r(t) - \bar{r}(t)\|_{L^2([-R_1, R_1])} \leq \epsilon_r$.

- The function $(t, y) \mapsto \partial_1 l(t, y)$ is approximated by $g(t, y)$ on $[-R_2, R_2]^2$:

$$g(t, y) = \sum_{m=1}^{M_2} \beta_m^2 \sigma(\langle \mathbf{a}_m^2, [t; y; 1] \rangle) \quad \text{with} \quad M_2 \leq \mathcal{O}(\epsilon_l^{-2})$$

  such that $\|g(t, y) - \partial_1 l(t, y)\|_{L^2([-R_2, R_2]^2)} \leq \epsilon_l$.

- The function $(s, t) \mapsto s \cdot r'(t)$ is approximated by $P(s, t)$ on $[-R_3, R_3]^2$:

$$P(s, t) = \sum_{m=1}^{M_3} \beta_m^3 \sigma(\langle \mathbf{a}_m^3, [s; t; 1] \rangle) \quad \text{with} \quad M_3 \leq \mathcal{O}(\epsilon_p^{-2})$$

  such that $\|P(s, t) - s \cdot r'(t)\|_{L^2([-R_2, R_2]^2)} \leq \epsilon_p$.

- The function $(u, v) \mapsto u \cdot v$ is approximated by $Q(u, v)$ on $[-R_4, R_4]^2$:

$$Q(u, v) = \sum_{m=1}^{M_4} \beta_m^4 \sigma(\langle \mathbf{a}_m^4, [u; v; 1] \rangle) \quad \text{with} \quad M_4 \leq \mathcal{O}(\epsilon_q^{-2})$$

  such that $\|Q(u, v) - u \cdot v\|_{L^2([-R_4, R_4]^2)} \leq \epsilon_q$.

for $i \in [n-1], j \in [K_i]$ and

$$\nabla_{u_k} \text{pred}(\mathbf{x}; \mathbf{w}) = r(\mathbf{v}_{n-1,k}^\top r(\cdots))$$

We'll later show that it is this difference in gradient that mainly contributes to the growth of transformer layers required. $n-2$ **attention only layers:** In the first attention-only layer, the transformer maps

$$\mathbf{h}_i = [\mathbf{x}_i; y_i; \mathbf{w}; \mathbf{0}; 1; t_i] \mapsto \mathbf{h}_i = [\mathbf{x}_i; y_i; \mathbf{w}; \mathbf{W}^{(2)}(\bar{r}(\mathbf{W}^{(1)}\mathbf{x})); \mathbf{0}; 1; t_i],$$

in the second attention-only layer, the transformer maps

$$\mathbf{h}_i^{(1)} \mapsto \mathbf{h}_i^{(2)},$$

where

$$\mathbf{h}_i^{(1)} = [\mathbf{x}_i; y_i; \mathbf{w}; \mathbf{W}^{(2)}(\bar{r}(\mathbf{W}^{(1)}\mathbf{x})); \mathbf{0}; 1; t_i],$$

and

$$\mathbf{h}_i^{(2)} = [\mathbf{x}_i; y_i; \mathbf{w}; \mathbf{W}^{(3)}(\bar{r}(\mathbf{W}^{(2)}(\bar{r}(\mathbf{W}^{(1)}\mathbf{x})))); \mathbf{0}; 1; t_i].$$

In the $p$-th layer, the transformer maps

$$\mathbf{h}_i^{(p-1)} \mapsto \mathbf{h}_i^{(p)},$$

where

$$\mathbf{h}_i^{(p-1)} = [\mathbf{x}_i; y_i; \mathbf{w}; \mathbf{W}^{(2)}(\bar{r}(\mathbf{W}^{(1)}\mathbf{x})); \cdots; \mathbf{W}^{(p)}(\bar{r}(\mathbf{W}^{(p-1)} \cdots \bar{r}(\mathbf{W}^{(1)}\mathbf{x}))); \mathbf{0}; 1; t_i],$$

and

$$\mathbf{h}_i^{(p)} = [\mathbf{x}_i; y_i; \mathbf{w}; \mathbf{W}^{(2)}(\bar{r}(\mathbf{W}^{(1)}\mathbf{x})); \cdots; \mathbf{W}^{(p+1)}(\bar{r}(\mathbf{W}^{(p)} \cdots \bar{r}(\mathbf{W}^{(1)}\mathbf{x}))); \mathbf{0}; 1; t_i].$$

We then prove why a transformer can achieve this mapping taking the $p$-th layer as an example.

We denote the $k$-th row of $\mathbf{W}^{(p)}(\bar{r}(\mathbf{W}^{(p-1)} \cdots \bar{r}(\mathbf{W}^{(1)}\mathbf{x})))$ as $\overline{\text{pred}}_{p,k}(\mathbf{x})$, which is a number. Consider the matrices $\{\mathbf{Q}_{k',k,m}^{(p)}, \mathbf{K}_{k',k,m}^{(p)}, \mathbf{V}_{k',k,m}^{(p)}\}_{k' \in [K_{p+1}], k \in [K_p], m \in [M_1]}$ so that for all $i, j \in N + 1$, we have

$$\mathbf{Q}_{k',k,m}^{(p)}\mathbf{h}_i = \begin{bmatrix} \mathbf{a}_m^1[1] \\ \mathbf{a}_m^1[2] \\ \mathbf{0} \end{bmatrix}, \quad \mathbf{K}_{k',k,m}^{(p)}\mathbf{h}_j = \begin{bmatrix} \overline{\text{pred}}_{p,k}(\mathbf{x}) \\ 1 \\ \mathbf{0} \end{bmatrix}, \quad \mathbf{V}_{k',k,m}^{(p)}\mathbf{h}_j = \beta_m^1 \cdot \mathbf{v}_{p+1,k'}[k]\mathbf{e}_{D_{k'}}.$$

Here $D_{k'}$ denotes the hidden space in the column vector $\mathbf{h}_i$ where the sum below is stored. Then we compute the update on the column $h_i$:

$$\sum_{m \in [M_1], k \in [K]} \sigma(\langle \mathbf{Q}_{k',k,m}^{(p)}\mathbf{h}_i, \mathbf{K}_{k',k,m}^{(p)}\mathbf{h}_j \rangle)\mathbf{V}_{k',k,m}^{(p)}\mathbf{h}_j = \sum_{k=1}^{K_p} \mathbf{v}_{p+1,k'}[k]\bar{r}(\overline{\text{pred}}_{p,k}(\mathbf{x})) \cdot \mathbf{e}_{D_{k'}}$$

The $n-1$-th layer follows the protocol of the mapping of the previous layers in the self-attention sub-layer. Thus after the $n-1$-th attention sub-layer, we have $\overline{\text{pred}}_n(\mathbf{x}; \mathbf{w})$ stored in the output column $\mathbf{h}_i^{(n-1.5)}$.

**The MLP sub-layer of the $n-1$-th layer:** In this feed forward layer we pick matrices $\mathbf{W}_1, \mathbf{W}_2$ such that $\mathbf{W}_1$ maps

$$\mathbf{W}_1\mathbf{h}_i^{(0.5)} = [\mathbf{a}_m^2[1] \cdot \overline{\text{pred}}(\mathbf{x}_i; \mathbf{w}) + \mathbf{a}_m^2[2] \cdot y_i' + \mathbf{a}_m^2[3] - R_2(1 - t_i)]_{m \in [M_2]},$$

and the entries of $\mathbf{W}_2$ are $(\mathbf{W}_2)_{(j,m)} = \beta_m^2 1\{j = D_0 + 2\}$. So

$$\mathbf{W}_2\sigma(\mathbf{W}_1\mathbf{h}_i^{(n-1.5)}) = \sum_{m \in [M_3]} \sigma(\langle \beta_m^2, [\overline{\text{pred}}(\mathbf{x}; \mathbf{w}); y_i'; 1] \rangle - R_2(1 - t_j)) \cdot c_m^2 \mathbf{e}_{D_0+2}$$

$$= 1\{t_j = 1\} \cdot g(\overline{\text{pred}}(\mathbf{x}_i; \mathbf{w}), y_i') \cdot \mathbf{e}_{D_0+2}.$$

So if we abbreviate $g_i = 1\{t_j = 1\} \cdot g(\overline{\text{pred}}(\mathbf{x}_i; \mathbf{w}), y_i')$, we get the output of this sublayer is

$$\mathbf{h}_i^{(n-1)} = [\mathbf{x}_i; y_i; \mathbf{w}; \mathbf{W}^{(2)}(\bar{r}(\mathbf{W}^{(1)}\mathbf{x})); \cdots; \mathbf{W}^{(n)}(\bar{r}(\mathbf{W}^{(n-1)} \cdots \bar{r}(\mathbf{W}^{(1)}\mathbf{x}))); g_i; \mathbf{0}; 1; t_i]$$

**Other layers to compute the approximate gradient:** Now, we look at the gradient of the neural networks again:

$$\nabla_{\mathbf{v}_{i,j}}\text{pred}(\mathbf{x}; \mathbf{w}) = \sum_{k=1}^{K_n} u_k r'(\mathbf{v}_{n-1,k}^\top r(\mathbf{W}^{(n-2)}r(\cdots)))\nabla_{\mathbf{v}_{i,j}}\mathbf{v}_{n-1,k}^\top r(\mathbf{W}^{(n-2)}r(\cdots))$$

for $i \in [n-1], j \in [K_i]$ and

$$\nabla_{u_k}\mathrm{pred}(\mathbf{x}; \mathbf{w}) = r(\mathbf{v}_{n-1,k}^\top r(\cdots)).$$

We observe that the term $\nabla_{\mathbf{v}_{i,j}}\mathbf{v}_{n-1,k}^\top r(\mathbf{W}^{(n-2)}r(\cdots))$ is in fact the gradient of an $n-1$ layer neural network, and a transformer needs $a_{n-1}$ layers to compute and store these gradients for all $i \in [n-1], j \in [K_i]$. After these $a_{n-1}$ layers, the approximate gradients are already stored in the hidden space of $\mathbf{h}_i$, and we denote the approximation of $\nabla_{\mathbf{v}_{i,j}}\mathbf{v}_{n-1,k}^\top r(\mathbf{W}^{(n-2)}r(\cdots))$ as $s_{i,j}^{(n-1)}$. Now consider the matrices $\{\mathbf{Q}_{k,m}, \mathbf{K}_{k,m}, \mathbf{V}_{k,m}\}_{k\in[K_n],m\in M_3}$ so that for all $i,j \in [N+1]$ we have

$$\mathbf{Q}_{k,m}\mathbf{h}_i = \begin{bmatrix} \mathbf{a}_m^3[1] \\ \mathbf{a}_m^3[2] \\ \mathbf{a}_m^3[3] \\ \mathbf{0} \end{bmatrix}, \quad \mathbf{K}_{k,m}\mathbf{h}_j = \begin{bmatrix} s_{i,j}^{(n-1)} \\ \overline{\mathrm{pred}}_{n-1}^k(\mathbf{x}_i; \mathbf{w}) \\ 1 \\ \mathbf{0} \end{bmatrix}, \quad \mathbf{V}_{k,m}\mathbf{h}_j = \beta_m^3 \cdot u_k \mathbf{e}_{D_{i,j}}.$$

Here $D_{i,j}$ denotes the place where we store the gradient of $\mathbf{v}_{i,j}$. A simple calculation yields

$$\sum_{m\in[M_1],k\in[K_n]} \sigma(\langle \mathbf{Q}_{k,m}\mathbf{h}_i, \mathbf{K}_{k,m}\mathbf{h}_j\rangle)\mathbf{V}_{k,m}\mathbf{h}_j = \sum_{k=1}^{K_n} u_k P(s_{i,j}^{(n-1)}, \overline{\mathrm{pred}}_{n-1}^k(\mathbf{x}_i; \mathbf{w})) \cdot \mathbf{e}_{D_{i,j}}$$

Now that all the terms that approximate $\nabla_{\mathbf{v}_{i,j}}\mathrm{pred}(\mathbf{x}; \mathbf{w})$ are stored in the hidden space (we denote them as $v_{i,j}$), we simply need another layer to compute the gradient of loss function.

Consider the matrices $\{\mathbf{Q}_{k,m}, \mathbf{K}_{k,m}, \mathbf{V}_{k,m}\}_{k\in[K_n],m\in M_4}$ so that for all $i,j \in [N+1]$ we have

$$\mathbf{Q}_{k,m}\mathbf{h}_i = \begin{bmatrix} \mathbf{a}_m^4[1] \cdot g_i \\ \mathbf{a}_m^4[2] \\ \mathbf{a}_m^4[3] \\ \mathbf{0} \end{bmatrix}, \quad \mathbf{K}_{k,m}\mathbf{h}_j = \begin{bmatrix} 1 \\ v_{i,j} \\ 1 \\ \mathbf{0} \end{bmatrix}, \quad \mathbf{V}_{k,m}\mathbf{h}_j = -\frac{(N+1)\eta\beta_m^4}{N} \cdot \mathbf{e}_{D_{i,j}}.$$

Now we get

$$\mathbf{g}(\mathbf{w}) := \frac{1}{N+1} \sum_{i=1}^{N+1} \sum_{(k,m)} \sigma(\langle \mathbf{Q}_{k,m}\mathbf{h}_i, \mathbf{K}_{k,m}\mathbf{h}_j\rangle)\mathbf{V}_{k,m}\mathbf{h}_j$$

$$= -\frac{\eta}{N} \sum_{i=1}^{N+1} \mathrm{approximate}(\partial_1 l(\mathrm{pred}(\mathbf{x}_i; \mathbf{w}), \mathbf{y}_i)) \cdot \mathrm{approximate}(\nabla_{\mathbf{w}}\mathrm{pred}(\mathbf{x}_i; \mathbf{w}))$$

Thus $\eta^{-1}\mathbf{g}(\mathbf{w})$ approximates $\nabla\hat{L}_N(\mathbf{w})$, and requiring $\|\eta^{-1}\mathbf{g}(\mathbf{w}) + \nabla\hat{L}_N(\mathbf{w})\|_{L^2} \le \epsilon$ yields the upper bound for the number of heads: $\mathcal{O}(nK^2\epsilon^{-2})$ and hidden dimension: $\mathcal{O}(nK^2\epsilon^{-2})$.

**Total number of layers:** From the analysis above, the total number of transformer layers required is $\mathcal{O}(a_n) = \mathcal{O}(a_{n-1}) + \mathcal{O}(n)$, and it is straightforward to check that $a_n$ is of order $\mathcal{O}(n^2)$. This completes the proof. $\qquad\square$