# OpenReview forum: "Transformers Perform In-Context Learning through Neural Networks"
_ICLR.cc/2024/Conference — ICLR 2024 Conference Withdrawn Submission_

### Official Review · Reviewer_m6gj · 2023-10-27

**Soundness:** 2 fair
**Presentation:** 2 fair
**Contribution:** 2 fair
**Rating:** 3
**Confidence:** 4

**Summary:**

This paper shows that transformers can approximate gradient descent on 2-layer and n-layer neural networks. It uses the capabilities of neural networks as a bridge to understand the in-context abilities of transformers

**Strengths:**

The expressive power of transformers for performing gradient descent on n-layer neural networks is a novel result.

**Weaknesses:**

1. The writing of the paper is very poor, and the paper's readability should be improved. Just to point out a bit,
in def 3, statements are introduced with many notations unspecified, e.g. Sobolev space, m, n;
2. The paper seems to have a strong connection with Bai et al, 2023, in. notations and results, their result naturally extends to the multi-layer setting. The novelty and contribution of the paper compared to Bai et al, 2023 is not clear to me.
3. The methodology of this paper is to first connect the transformer's expressive ability and neural networks, and then use the well-established approximation ability of neural networks, both of these two parts look not novel enough.



[1] Transformers as Statisticians: Provable In-Context Learning with In-Context Algorithm Selection. Yu Bai, Fan Chen, Huan Wang, Caiming Xiong, Song Mei, 2023

**Questions:**

See weakness

---

> ### Author Response · Authors · 2023-11-21
>
> Thank you for your valuable comments and suggestions. We provide our response below.
>
> **The novelty compared to Bai et al, 2023**
>
> Apart from generalizing the 2-layer NN setting to the n-layer setting, we actually used a different approximation theorem to achieve a better error bound for our results. It leads to the decrease of number of heads and hidden dimension from $O(\log\frac{1}{\epsilon}\epsilon^{-2})$ in [1] to $O(\epsilon^{-2})$ in our result.

---

### Official Review · Reviewer_Ah1w · 2023-10-30

**Soundness:** 2 fair
**Presentation:** 2 fair
**Contribution:** 2 fair
**Rating:** 3
**Confidence:** 4

**Summary:**

This paper studies how transformers approximate n-layer neural networks by gradient descent and its variants. The theoretical results characterize the required number of heads, hidden dimensions, and layers of transformers. The authors also provide a comparison between deep, narrow networks with shallow, wide networks in terms of computational resources.


----------------------------

**After rebuttal**: Thank you for your response, and sorry for the late reply. I am sorry that I will keep the rating of 3. There are several reasons. (1) Weaknesses 1, 2, and 3 are not answered. (2) It is not a good reason to say other papers don't do experiments. First, I always feel that no experiment is a disadvantage. Second, I think other papers without experiments have their own originality and contributions such that they get accepted. I am not satisfied with the contributions in this paper, so experiments could be a complement.

**Strengths:**

1. The problem to be solved is significant and interesting to the community.
2. This work extends the analysis to deep neural networks and makes a comparison between shallow networks and deep networks.

**Weaknesses:**

1. The biggest concern is the technical contributions of this work. I treat this work as a follow-up work of [Bai et al., 2023]. Therefore, it is very important to state the contribution beyond this existing work. I am not sure of the difficulty and the novelty of proving Theorem 2.

2. This work lacks empirical justification.

3. The complete proof of Theorem 1,3,4 are not provided

[Bai et al., 2023]: Transformers as statisticians: Provable in-context learning with in-context algorithm selection.

**Questions:**

1. Can you show the comparison between deep, narrow networks and shallow, wide networks by experiments?

2. What is the definition of $\mathcal{W}$, the domain of $\bf{w}$? I do feel Assumption 2 is too strong. Is the existence of the MLP layer in Assumption 2 provable rather than assumed?

---

> ### Author Response · Authors · 2023-11-21
>
> Thank you for your valuable comments and suggestions. We provide our response to each question below.
>
> **Question 1. Experimental results.**
>
> So far many works ([1], [2], [3]) have theoretically proven the advantage of deep networks over shallow networks, but there lacks experimental results. Empirical validation still needs further investigation.
>
> **Question 2. Assumption 2**
>
> The domain $\mathcal{W}$ of $w$ is a compact set in $\mathbb{R}^d$, meaning its bounded and closed. Here $d$ is the dimension of $w$. The existence of such an MLP layer requires some construction, but it is reasonable to assume the existence since an MLP layer projects the input into another space.
>
> [1] Ronen Eldan and Ohad Shamir. The power of depth for feedforward neural networks. In Conference on learning
> theory, pages 907–940. PMLR, 2016.
>
> [2] Shiyu Liang and Rayadurgam Srikant. Why deep neural networks for function approximation? arXiv preprint
> arXiv:1610.04161, 2016.
>
> [3] Itay Safran and Ohad Shamir. Depth-width tradeoffs in approximating natural functions with neural networks.
> In International conference on machine learning, pages 2979–2987. PMLR, 2017.

---

### Official Review · Reviewer_SDBk · 2023-11-05

**Soundness:** 3 good
**Presentation:** 2 fair
**Contribution:** 2 fair
**Rating:** 3
**Confidence:** 3

**Summary:**

This paper studies the in-context learning capabilities of Transformer-based models. It is shown by construction that Transformer can approximately implement gradient descent steps on the parameters of certain neural networks, where the upper bounds on the required number of heads, hidden size, and number of layers are provided. This suggests that Transformer can perform in-context learning by approximating a neural network.

**Strengths:**

The in-context learning capability of Transformer is an important topic recently in the community. This paper provides an interesting perspective of explaining the in-context learning performance of Transformer. The authors have clearly explained their idea, and the presentation is easy to follow.

**Weaknesses:**

- The main result presents the existence of the weights that enable Transformer to do gradient descent on a certain neural network. It is not shown if Transformer can be actually trained to do so.
- Theorem 3 and Theorem 4 only provide upper bounds, which do not necessarily suggest a separation.
- It seems that many recent papers on in-context learning of Transformer are missing. One of the most related ones is
    - Trainable transformer in transformer. Panigrahi, Abhishek and Malladi, Sadhika and Xia, Mengzhou and Arora, Sanjeev.
- There are no numerical experiments supporting the theoretical results.
- Typo: Under Definition 2, "We doesn't" -> "We do not"

**Questions:**

1. In Definition 3, what is the definition of $W_{loc}^{m,\infty}(\mathbb{R})$?
2. It is not clear to me what Assumption 2 means. Is there a concrete example?
3. In Assumption 1, it is not clear to me what it suggests for the loss function to have finite Barron norm. According to the definition, it seems that we then need $\ell(w) \to 0$ as $\|w\| \to \infty$. Does the commonly-used $\ell_2$ loss satisfy this?
4. I find the results in Section 4 a bit confusing. Specifically, what does it mean by "learn the neural network"? Is it about doing gradient descent on certain loss? Otherwise, how is this related to the results in Sections?

---

> ### Author Response · Authors · 2023-11-21
>
> Thank you for your valuable comments and suggestions. We provide our response to each question below.
>
> **Question1. The definition of $W_{loc}^{m,\infty}(\mathbb{R})$**
>
> $W^{m,\infty}(\mathbb{R})$ denotes the Sobolev space that is a subset of the $L^\infty$ space. For any $f\in W^{m,\infty}$ we have $f$ is continuously differentiable up to order $m$, and these derivatives have finite $L^\infty$ norm. Here $W_{loc}^{m,\infty}(\mathbb{R})$ is equivalent to $W^{m,\infty}(\Omega)$, where $\Omega\subset \mathbb{R}$ is a bounded domain.
>
> **Question 2. Meaning of assumption 2**
>
> This assumption follows Bai et al, 2023 in Theorem G.1. Since $\mathcal{W}$ is a bounded domain, we assume that an MLP layer can project $w$ into its domain for simplicity, and whether there's a concrete example remains requires some construction, but it is reasonable to assume that such a layer exists due to the essence of the MLP layer, which projects the input into another space.
>
> **Question 3. Functions having finite Barron norm**
>
> We would like to refer the reviewer to section IX in [1] where Barron discussed functions that have finite Barron norm. These functions include sigmoidal functions, radial functions, linear functions and polynomials, and most importantly, functions of sufficiently high order, with partial derivatives of $f(x)$ of order $s=[d/2]+2$ continuous on $\mathbb{R}^d$. So the commonly used $l_2$ loss function clearly falls into this category.
>
> **Question 4. Results in section 4**
>
> First we know that a transformer can learn neural networks in-context, meaning given the input, a transformer can give the output of the NN, and what we mean by "learn the neural network" is based on this fact. So our takeaway message is "When a transformer learns a function in-context, it actually learns the neural network that approximates the function in-context".
>
> [1] Andrew R Barron. Universal approximation bounds for superpositions of a sigmoidal function. IEEE Transactions on Information theory, 39(3):930–945, 1993.

---

> > ### Comment · Reviewer_SDBk · 2023-11-23
> > **Reply to rebuttal**
> >
> > I thank the authors for the response.
> >
> > If I understand it correctly, in the definition of Barron norm, $\hat f$ refers to the Fourier transform of $f$. It would be helpful to clarify this notation.
> >
> > Theorem 2 in Section 3 states that Transformer can perform one step of gradient descent for a neural network. It's not immediately clear to me how this implies that Transformer can learn the target neural network as claimed in Theorem 3, and the proof of Theorem 3 is indeed based on Theorem 2.